# Composition Design and Tensile Properties of Additive Manufactured Low Density Hf-Nb-Ta-Ti-Zr High Entropy Alloys Based on Atomic Simulations

**DOI:** 10.3390/ma16114039

**Published:** 2023-05-29

**Authors:** Zhuoheng Liang, Yiming Wu, Yu Miao, Wei Pan, Yongzhong Zhang

**Affiliations:** 1GRINM Group Corporation Limited, National Engineering & Technology Research Center for Non-Ferrous Metals Composites, Beijing 101407, China; lzh_dayu1995@163.com (Z.L.);; 2GRINM Metal Composites Technology Co., Ltd., Beijing 101407, China; 3General Research Institute for Nonferrous Metals, Beijing 100088, China; 4School of Materials Science and Engineering, Harbin Institute of Technology, Harbin 150001, China; 5College of Computer Science and Technology, Harbin Engineering University, Harbin 150001, China

**Keywords:** high-entropy alloy, tensile mechanical properties, atomic simulations, laser melting deposition

## Abstract

High-entropy alloy (HEA) is a new type of multi-principal alloy material and the Hf-Nb-Ta-Ti-Zr HEAs have attracted more and more attention from researchers due to their high melting point, special plasticity, and excellent corrosion resistance. In this paper, in order to reduce the density of the alloy and maintain the strength of the Hf-Nb-Ta-Ti-Zr HEAs, the effects of high-density elements Hf and Ta on the properties of HEAs were explored for the first time based on molecular dynamics simulations. A low-density and high-strength Hf_0.25_NbTa_0.25_TiZr HEA suitable for laser melting deposition was designed and formed. Studies have shown that the decrease in the proportion of Ta element reduces the strength of HEA, while the decrease in Hf element increases the strength of HEA. The simultaneous decrease in the ratio of Hf and Ta elements reduces the elastic modulus and strength of HEA and leads to the coarsening of the alloy microstructure. The application of laser melting deposition (LMD) technology refines the grains and effectively solves the coarsening problem. Compared with the as-cast state, the as-deposited Hf_0.25_NbTa_0.25_TiZr HEA obtained by LMD forming has obvious grain refinement (from 300 μm to 20–80 μm). At the same time, compared with the as-cast Hf_0.25_NbTa_0.25_TiZr HEA (σ_s_ = 730 ± 23 MPa), the as-deposited Hf_0.25_NbTa_0.25_TiZr HEA has higher strength (σ_s_ = 925 ± 9 MPa), which is similar to the as-cast equiatomic ratio HfNbTaTiZr HEA (σ_s_ = 970 ± 15 MPa).

## 1. Introduction

High-entropy alloy (HEA) is a new type of multi-principal alloy containing at least 5 elements with concentrations varying from 5 to 35 atom percentage [1,2]. Due to the completely new composition characteristics different from traditional alloys, high-entropy alloys have various special properties in terms of thermodynamics and lattice structure: high entropy mainly due to configurational entropy, lattice distortion due to large variation in atomic sizes of constituent elements and mutual chemical bonds, sluggish diffusion kinetics due to lattice distortions and overall properties due to diverse multi-level microstructure [3,4]. Based on the special atomic structure, HEAs tend to have high strength and hardness [5,6], excellent wear resistance and corrosion resistance [7,8].

As a single-phase BCC alloy, Hf-Nb-Ta-Ti-Zr HEAs have special plasticity characteristics and have been widely studied [9]. Liu et al. [10] explored the tensile creep process of HfNbTaTiZr HEA and determined the dominant role of Ta element in this process. The fracture toughness mechanism of HfNbTaTiZr HEA was studied by Fan et al. [11], and the fracture toughness KJIC of HEA was determined to be 210 MPa m1/2 by the “single sample” flexibility method which renders this HEA among the toughest metallic materials. Yang et al. [12] explored the spontaneous passivation phenomenon of equimolar HfNbTaTiZr HEA in 310 K Hank solution, and explored the application of this HEA in the field of biomedical materials (Bio-HEAs). However, the large viscosity of the melt of HEAs has caused difficulties in its forming, which limits the practical application of HEAs [13].

Laser melting deposition (LMD) has been established as a versatile additive manufacturing (AM) technology, which has obvious advantages in forming complex shapes [14,15]. Due to the thermal field characteristics different from traditional casting methods, additive manufacturing technology has unique advantages in the control of alloy microstructure [16,17]. The characteristics of LMD make it able to effectively solve the problems faced by HEAs forming [18,19]. However, due to the fast cooling rate and large temperature gradient in the LMD process, there will be uneven residual stress in the as-deposited alloy obtained by forming [20]. Therefore, it is necessary to study and control the alloy composition to make it suitable for LMD forming.

In addition, the researchers try to reduce the density of Hf-Nb-Ta-Ti-Zr HEAs to make it suitable for a wider field [21]. In order to provide theoretical support for composition proportion adjustment, the influence of different elements on alloy properties needs to be explored. Molecular dynamics (MD) simulation is a method of simulating the physical motion trajectories and states of atoms and molecules based on the principles of Newtonian mechanics, which can provide references and supplements to experiments by studying the structure and dynamic behavior of materials at the atomic scale [22,23]. Increasingly researchers have applied MD simulation to the research and development of HEAs composition, such as AlCrCoCuFeNi [24], NbMoTaW [25], HfNbTaZr [26] and CoFeNiTi [27] alloys. Due to the special multi-principal composition of HEA, the analysis of its properties and structure is complicated. The performance of HEAs can be quickly and intuitively explored and evaluated using MD simulations. There are reasons to believe that the composition design based on MD simulation, combined with the forming method of LMD with fast response capability, can effectively improve the design efficiency of HEAs and promote the engineering application of HEAs. In this paper, MD simulation method is used to explore the impact of high-density Hf and Ta elements on the performance of Hf-Nb-Ta-Ti-Zr HEA, and then LMD method is used to shape it to obtain a paradigm for low-density HEAs composition design.

## 2. Materials and Methods

### 2.1. MD Model and Method

In this study, LAMMPS (version 3 2020) [28], a molecular dynamics simulation software, was used to realize the MD simulation calculations of Hf_x_NbTaTiZr (x = 0.25, 0.5, 0.75, 1), HfNbTa_x_TiZr (x = 0.25, 0.5, 0.75, 1) and Hf_x_NbTaxTiZr (x = 0.25, 0.5, 0.75, 1) HEAs. When constructing the alloy model by LAMMPS, the atoms of the five elements are randomly distributed in a BCC deconstructed atomic box, and the way of random distribution is defined by the random seed value, which can be any positive integer [29]. In order to make the simulation results as close as possible to the actual situation of the as-cast alloy, a thousand models were established for each composition scheme by changing the random seed, and alternately use the style cg, the Polak-Ribiere version of the conjugate gradient (CG) algorithm and the style sd—a steepest descent algorithm—to minimize energy and optimize geometry structure [30,31]. Calculate the energy of each optimized model and select the lowest one for subsequent MD simulations. The entire iterative modeling and screening process is shown in Figure 1a. The atomic interactions of the Hf-Nb-Ta-Zr were described by many-body embedded-atom potential [32]. The obtained stable HfNbTaTiZr HEA models of different components are shown in Figure 1b. As shown in Figure 1b, there are 41,472 atoms in the models, with the size of the models being 24a × 24a × 36a (a is the lattice constant).

During the molecular dynamics simulation, the periodic boundary conditions have been used in x-, y-, z- directions of the models. Before performing mechanical performance simulations, the entire models underwent full relaxation for the isothermal and isostatic pressure (NPT) system, which enables the model to simulate the melting-cooling process and improves the reliability of the simulation results. For the simulation process of tensile mechanical properties, the time-step was set to be 0.001 ps and the Verlet leapfrog method was used to solve the MD integral equation. During the loading process, the pressure in the other two directions was always set to zero except for the z-direction where the deformation was applied. Under the NPT system, the model loading process was realized by using a strain rate of 4 × 108 s − 1 in the simulations.

The atom type was determined by common neighbor analysis (CNA) [33]. The software OVITO (version 3.8.4) [34] was used for the visualization of the models and the simulation processes performed on them, as well as for the structural analysis and image processing. The dislocation extraction algorithm (DXA) was employed to analyze the dislocation evolution [35].

### 2.2. Experimental Materials and Methods

To verify the reliability of the MD simulation results, as-cast samples of alloys of Hf_x_NbTa_x_TiZr (x = 0.25, 0.50, 0.75, 1) were prepared and characterized and compared with the simulation results. Experimental as-cast alloys were prepared in an arc melting furnace, and raw materials were high-purity (≥99.9%) Hf, Nb, Ta, Ti and Zr metals. In order to improve the uniformity of the composition and prevent composition segregation, each alloy ingot has been turned over and remelted 4–6 times during the melting process and oblate as-cast alloy ingots with a maximum diameter of about 49 mm and a thickness of about 14 mm were finally obtained. The alloy ingots were cut at the maximum diameter position to prepare the tensile mechanical properties experimental samples. 

The LMD technology was used to verify the feasibility obtained by MD simulation of applying the HEAs composition to additive manufacturing and the composition of Hf_0.25_NbTa_0.25_TiZr was chosen for its low density, excellent plasticity, and acceptable strength reduction. In this paper, the Hf_0.25_NbTa_0.25_TiZr HEAs ingots were firstly prepared by vacuum electromagnetic levitation melting technology and the raw materials were high-purity (≥99.9%) Hf, Nb, Ta, Ti and Zr metals. Using graphite dies, HEAs were remelted into Ø30 mm × 250 mm cylindrical ingots which would be used for the next step of plasma rotation electrode process (PREP). Prepared HEA powder for laser additive manufacturing with a particle size of 30–150 μm (D10 = 45.23 μm, D50 = 72.38 μm, D90 = 116.4 μm) has good sphericity. The morphology and particle size statistic of the powder are shown in Figure 2.

The thin-wall as-deposited samples of Hf_0.25_NbTa_0.25_TiZr alloy with the size of 50 mm (length) × 12 mm (height) × 3 mm (width) were prepared with a self-assembling laser melting deposition (LMD) process system, which consisted of a 3 kW semiconductor laser with the wavelength of 900–1070 nm, a coaxial powder feeding nozzle, a DPSF-2 powder feeder with two hoppers, a computer numerical control (CNC) platform and a glove box filled with argon. Figure 3a displays the schematic diagram of the LMD process and the deposition path. The LMD process was conducted using laser power of 600 W, 800 W, 1000 W, 1200 W and 1400 W, scanning rate of 4 mm/s, powder feeding rate of 8 g/min and height increment of 0.5 mm of each layer. As-deposited samples prepared with five different process parameters is shown in Figure 3b and all these samples were measured and evaluated using a Vickers hardness tester. The as-deposited alloy sample with better hardness was selected for tensile mechanical test.

The X-ray diffractometer (XRD, SmartLab 9kW, Rigaku, Japan) was used to analyze the phase structure of the surface of the samples. The scanning angle was 10~90° 2θ, and the scanning speed was 10°/min. Morphology and microstructure of alloy powders and samples of as-cast and as-deposited HEAs were observed using field-emission scanning electron microscopy (SEM, JSM-7900F JEOL Ltd., Tokyo, Japan). The electron backscatter diffraction (EBSD) equipped in the same electron microscope was used to observe and count the grain orientation and grain size of the as-cast and as-deposited HEA samples. Tensile mechanical performance test is carried out by NT100 electronic tensile machine at a constant displacement speed of 0.3 mm/min. The sampling method is shown in Figure 3c.

## 3. Results and Discussion

### 3.1. Effects of Hf and Ta Contents

In order to obtain a HEA composition that reduces the density of the alloy while retaining the strength as much as possible, this paper attempts to explore the effect of reducing the proportion of high-density elements, Hf and Ta, on the mechanical properties of HfNbTaTiZr HEAs. By individually adjusting the ratio of one element to conduct MD simulations, the effects of Hf and Ta elements on the mechanical properties of high-entropy alloys were investigated separately. The room temperature tensile stress–strain curves obtained by MD simulation are shown in Figure 4a,b. The yield strength values obtained from the tensile simulation of alloys with different compositions have been counted, and the variation trend of the yield strength (σ_0.2_) is shown in Figure 4c. It can be clearly observed that the yield strength of the alloy decreases with the decrease of the Ta element proportion, and on the other hand, increases with the increase of the Hf element proportion. This is in line with the experimental results of previous scholars who have explored the effect of different elements on the properties of the alloy by removing the elements in the Hf-Mo-Nb-Ta-Ti-Zr six-element HEAs, and found that with the absence of Ta element the yield strength and tensile strength of the alloy would decrease, and the elongation would increase [36]. On the other hand, with the decrease of Ta element content, the elastic modulus of the alloy also decreased significantly, and with the decrease of Hf content, the elastic modulus of the alloy has a slight increase which is not obvious. During the stretching process, as the grains of HfNbTaTiZr HEAs undergo deformation, the dislocations move inside the material, increasing the dislocation density [37,38]. Figure 5 shows a schematic diagram of the dislocation distribution inside the HfNbTaTiZr HEAs with elongation of 20 percent obtained from MD simulations. As shown in Figure 5, the dislocation density of the alloy decreased with the decrease of the atomic content of Ta element and increased with the decrease of the atomic content of Hf element, which is consistent with the results presented by the stress–strain curve obtained from the tensile simulation. The total length of dislocations in each HEA model was calculated using OVITO, and the results support the description of the trend of dislocation density shown in the schematic diagram.

Based on the existing simulation results, this paper reduces the atomic content of Ta and Hf at the same time to investigate their effects on the mechanical properties of the HEA, the tensile stress–strain curve obtained from the simulation is shown in Figure 4d. On the other hand, to further verify the accuracy of the simulation results, as-cast Hf_x_NbTa_x_TiZr HEA samples (x = 0.25, 0.5, 0.75, 1) were prepared and tested to compare with the simulation results. The tensile stress–strain curve obtained from the test is shown in Figure 4e. The tensile yield strengths of HEAs with different compositions were calculated based on the simulation curves and compared with the tensile strengths obtained by testing. The comparison results are shown in Figure 4f. It can be seen from the comparison results that the variation trend of the simulated values is in line with the actual test results. The variation trend of dislocation density shown in Figure 5 also provides evidence.

Although the trends are consistent, the differences between the actual stress–strain curves obtained from the tests are more pronounced. Therefore, the microstructure of the as-cast alloy samples was analyzed in this paper. As the grain reconstruction images shown in Figure 6 demonstrate, the reduction of Hf and Ta elements significantly coarsens the microstructure of Hf_x_NbTa_x_TiZr HEAs. This phenomenon explains why the difference in the simulated values of the properties of HEA with different components is smaller than the difference in the experimental values. On the other hand, this indicates that controlling the grain size is the key to the mechanical properties of Hf_x_NbTa_x_TiZr HEAs.

### 3.2. Preparation and Study of As-Deposited Samples

Based on previous MD simulation results, a composition of Hf_0.25_NbTa_0.25_TiZr was selected for the further exploration of as-deposited samples formed using LMD. During the additive manufacturing process, a high temperature gradient will be constructed, which will cause the accumulation of thermal stress inside the alloy [39]. Therefore, alloys suitable for AM forming should retain as much plasticity as possible based on high strength. In this paper, a preliminary exploration of process parameters was carried out to determine the feasibility of this composition being applied to LMD forming. The as-deposited thin-wall Hf_0.25_NbTa_0.25_TiZr alloy samples with smooth surface and no obvious internal defects were successfully prepared by LMD technology, which proved the feasibility of this alloy composition scheme being applied to laser additive manufacturing. Schematic of the thin-wall samples has been shown in Figure 3b and the number of reciprocating times of the laser scanning during the forming process of each sample is the same. The phase compositions of as-cast, powder and as-deposited samples were evaluated using XRD and the resulting diffraction patterns are shown in Figure 7. It can be clearly observed that the phase composition of the alloy at different stages of the LMD forming process are the same.

The hardness of the alloy samples was measured with a Vickers hardness tester, and the obtained hardness statistics are shown in Figure 8a. As shown in Figure 8a, the as-deposited samples formed at the laser power of 800–1200 W have higher hardness levels, and the samples formed by the laser power of 1200 W have the highest average hardness. The heights of the as-deposited samples were measured and the results were 105 mm (600 W), 120 mm (800 W), 135 mm (1000 W), 120 mm (1200 W) and 110 mm (1400 W). Since the process of forming the 5 samples is the same, the forming process with the laser power of 1000 W obviously has the highest forming efficiency. The grain size of the HEA samples was measured using EBSD and the resulting statistics are shown in Figure 8b–f. From the statistical results, compared with 600 W and 1400 W, the grain size of the alloy samples under 800 W–1200 W laser power is smaller, and grain size distribution range of 1200 W is the lowest, which is consistent with the hardness measurement results.

Since the properties of the as-deposited alloys obtained by the laser power of 800–1200 W are similar, the as-deposited alloy sample formed by the laser power of 1000 W was selected for further microstructure and property analysis for its highest deposition efficiency. As shown in Figure 9, the grains of the as-deposited alloy samples were observed by EBSD and compared with those of the as-cast samples. The grain diameter of the as-cast HEA is about 300 μm (Figure 9d), while the as-deposited HEA formed by LMD generally has a finer microstructure as shown in Figure 9b,c. Grain refinement can effectively improve the properties of the alloy [40], and the comparison of grain size effectively proves the regulation and optimization of LMD forming Hf_0.25_NbTa_0.25_TiZr HEA on its microstructure and properties. Observing the adjacent local areas of y–z cross section, as shown in Figure 9b,c, it is found that the as-deposited alloy structure presents an alternating distribution of coarse columnar grains and fine equiaxed grains along the z-direction. The alternating distribution occurs because the thermal history of each location of the as-deposited alloy during LMD forming consists of successive thermal cycles in different temperature ranges [41]. The thermal cycle generated by forming the new layer, combined with the lower nucleation barrier, causes the epitaxial growth of the grains of the already formed layer to form the columnar grains shown in Figure 9b [42,43]. However, at the same time, as shown in Figure 9a,c, the equiaxial crystals in the as-deposited HEA still occupy a large proportion, which effectively reduces the anisotropy of properties caused by the columnar crystals [44,45]. As shown in Figure 9e, an amount of dislocations (DS in Figure 9e) were found inside the as-deposited alloy grains. The existence of multiple dislocations not only proves the plasticity of the Hf_0.25_NbTa_0.25_TiZr HEA, but also proves that the laser additive manufacturing process can improve the mechanical properties of the HEA by increasing the density of dislocations through the accumulation of thermal stress.

The experimental mechanical properties of the obtained deposited Hf_0.25_NbTa_0.25_TiZr HEA samples were tested and compared with the cast samples, as shown in Figure 7b. It can be intuitively observed that compared to the as-cast one, the tensile strength and yield strength and plasticity of the as-deposited Hf_0.25_NbTa_0.25_TiZr HEA have significantly improved, which is close to the performance of the as-cast Hf_x_NbTa_x_TiZr (x = 1) HEA. Specific strength and elongation values are shown in Table 1. The effectiveness of the grain refinement in enhancing the strength of polycrystalline metals is known to be ascribed to the grain-boundary hardening, and can be quantified by the well-known Hall-Petch relation [46,47]:(1)$$\sigma_{y}=\sigma_{0}+k_{HP}D^{-1/2}$$
where $\sigma_{y}$ is the yield strength, *D* is the average grain size, $\sigma_{0}$ is a constant that can be considered as either the frictional stress induced by dislocation motions or internal back stresses $k_{HP}$ is a constant that can be viewed as an easure of the grain-boundary resistance to slip transfer [47]. Compared to the as-cast HEA, the changes in microstructure and properties of the as-deposited HEA conform to the description of the Hall-Petch formula.

## 4. Conclusions

In summary, molecular dynamics simulations were used to explore the effects of Hf and Ta elements on the performance of HfNbTaTiZr HEAs. Based on the results of molecular dynamics simulation, a low density Hf_0.25_NbTa_0.25_TiZr HEA suitable for laser melting deposition forming technology was determined. The problem of alloy coarsening was solved by LMD technology, and the balance between light weight and strength of HEA was finally realized. The simulation process and results, as well as HEAs with different composition ratios formed in different ways, were explored and tested. The results obtained are as follows:Through MD simulation, it is found that the strength of HEA decreases with the decrease of Ta content, while increases with the decrease of Hf content. This result is consistent with the experimental results of the existing research. The reliability of the simulation results has been preliminarily confirmed.The atomic ratios of Hf and Ta elements were adjusted synchronously, and their effects on the performance of HEA were simulated. The results show that as the ratio of the two elements decreases, the yield strength and elastic modulus of HEA decrease. A small amount of as-cast HEA samples with the same composition were prepared and tested. The variation trend of properties obtained by the test is consistent with the MD simulation results which further confirms the reference value of the MD simulation results.A low-density HEA composition was designed based on MD simulation results and successfully used in LMD forming. The as-deposited samples have good density, and the deposition efficiency first increases and then decreases with the increase of laser power.Compared with as-cast HEA, as-deposited HEA has finer grains and higher mechanical properties. Compared with the as-cast equiatomic ratio HfNbTaTiZr HEA, the as-deposited Hf_0.25_NbTa_0.25_TiZr HEA has lower density and similar mechanical properties. Pre-planned engineering goal has been successfully achieved.

MD simulation can qualitatively investigate the effect of composition on alloy properties, which greatly improves the efficiency of composition design. However, it can also be seen from the verification process that there is still a quantitative difference between it and the actual value. This is because the scale of the model used in the MD simulation is too small to accurately reflect the effect of grain boundaries and grain size on the properties of the alloy, so it needs the assistance of experiments when applied to composition design. However, the accuracy of qualitative analysis reduces the amount of experiments in the process of component design. Combined with the additive manufacturing technology with fast response characteristics, the component design based on MD simulation has a good application prospect.

## Figures and Tables

**Figure 1 materials-16-04039-f001:**
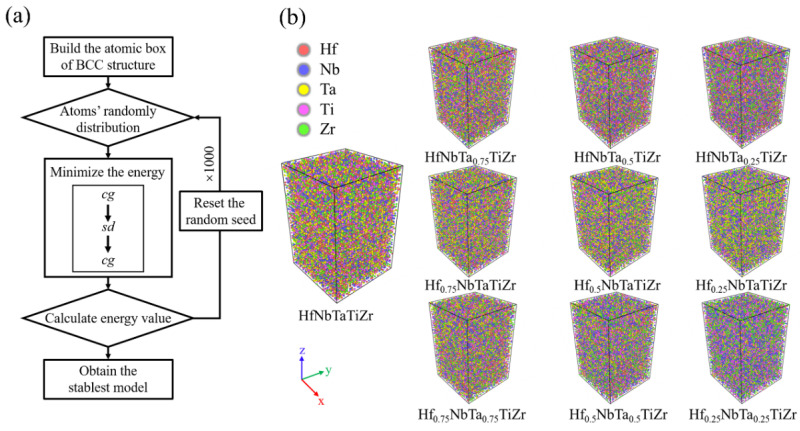
(**a**) Diagram of repeated modeling and screening, (**b**) Hf_x_NbTaTiZr HfNbTa_x_TiZr and HfxNbTa_x_TiZr HEA models with different Hf and Ta contents for x = 0.25, 0.5, 0.75, 1.00.

**Figure 2 materials-16-04039-f002:**
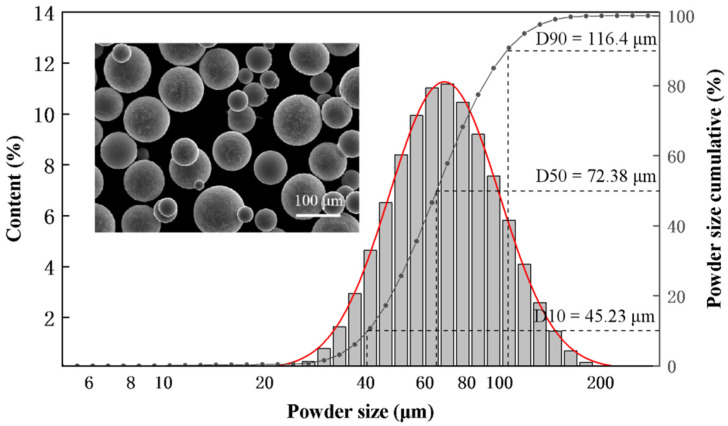
Morphology and particle size statistic of Hf_0.25_NbTa_0.25_TiZr HEA powder.

**Figure 3 materials-16-04039-f003:**
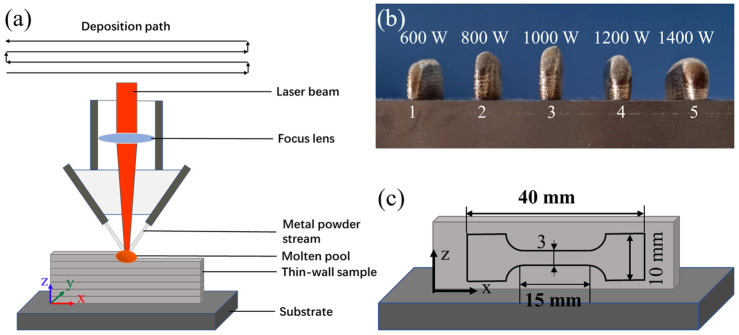
(**a**) Schematic diagram of LAM process and deposition path, (**b**) as-deposited samples prepared with different process parameters, (**c**) tensile samples sampling method.

**Figure 4 materials-16-04039-f004:**
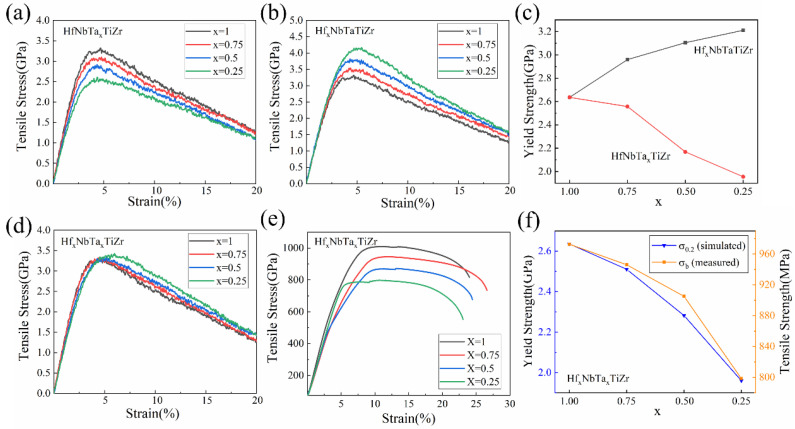
Results and comparison of tensile mechanics MD simulations and tests at room temperature: (**a**) MD simulation of the effect of Ta content, (**b**) MD simulation of the effect of Hf content, (**c**) variation trend of the yield strength, (**d**) room temperature tensile MD simulation result of Hf_x_NbTa_x_TiZr, (**e**) room temperature tensile tests result of Hf_x_NbTa_x_TiZr, (**f**) variation trend of the strength.

**Figure 5 materials-16-04039-f005:**
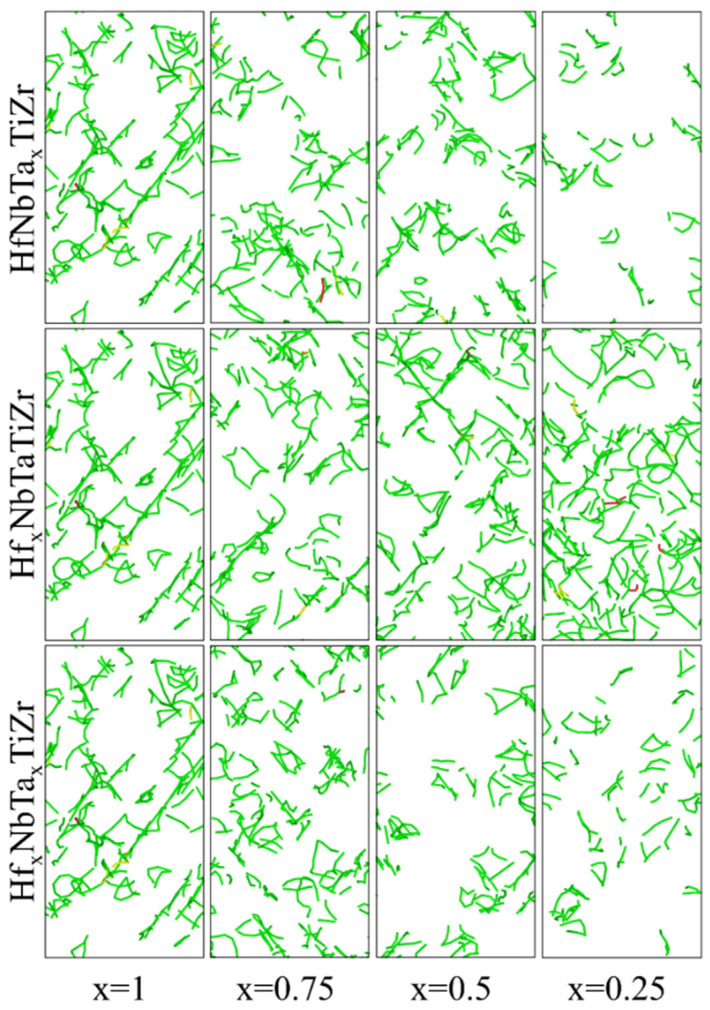
Schematic diagrams of dislocation density obtained from MD simulations.

**Figure 6 materials-16-04039-f006:**
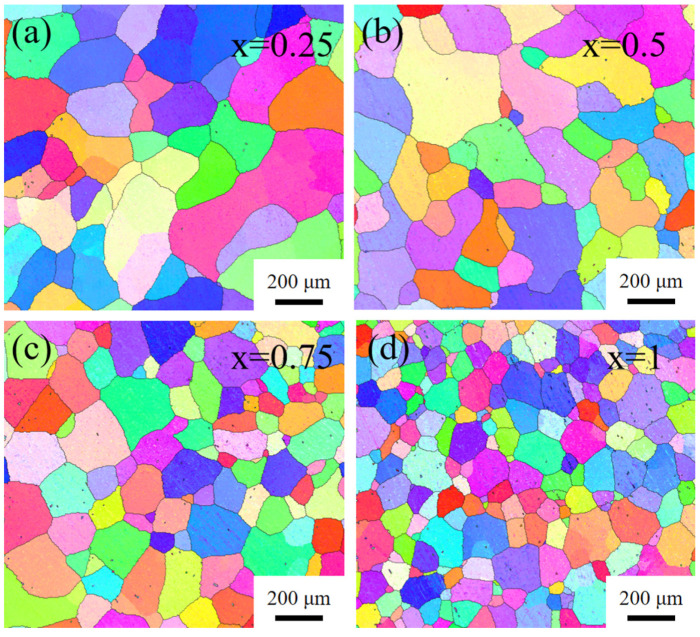
Grain reconstruction images of as-cast Hf_x_NbTa_x_TiZr HEAs: (**a**) x = 0.25, (**b**) x = 0.5, (**c**) x = 0.75, (**d**) x = 1.

**Figure 7 materials-16-04039-f007:**
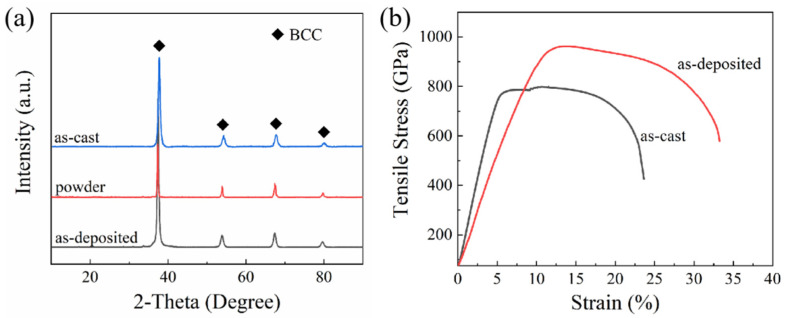
(**a**) XRD patterns of the as-cast, powder and as-deposited Hf_0.25_NbTa_0.25_TiZr HEA, (**b**) Stress–strain tensile characteristics of the as-cast and as-deposited samples.

**Figure 8 materials-16-04039-f008:**
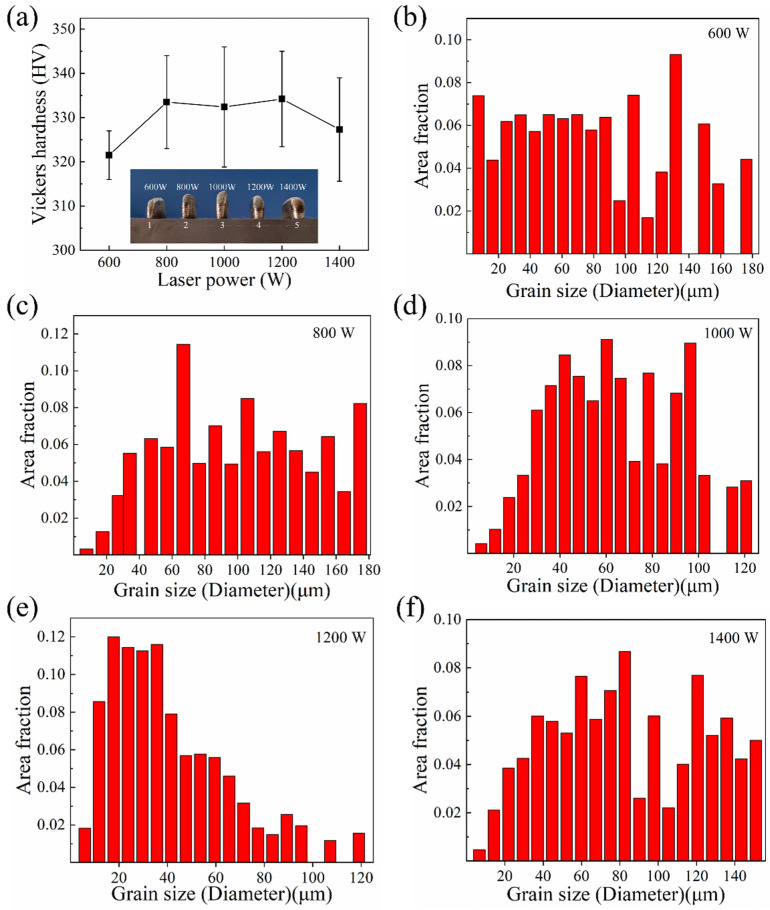
Properties and grain size of as-deposited Hf_0.25_NbTa_0.25_TiZr HEA samples obtained by different processes: (**a**) as-deposited samples’ photo and Vickers hardness, (**b**–**f**) grain size of samples formed with laser power of (**b**) 600 W, (**c**) 800 W, (**d**) 1000 W, (**e**) 1200 W and (**f**) 1400 W.

**Figure 9 materials-16-04039-f009:**
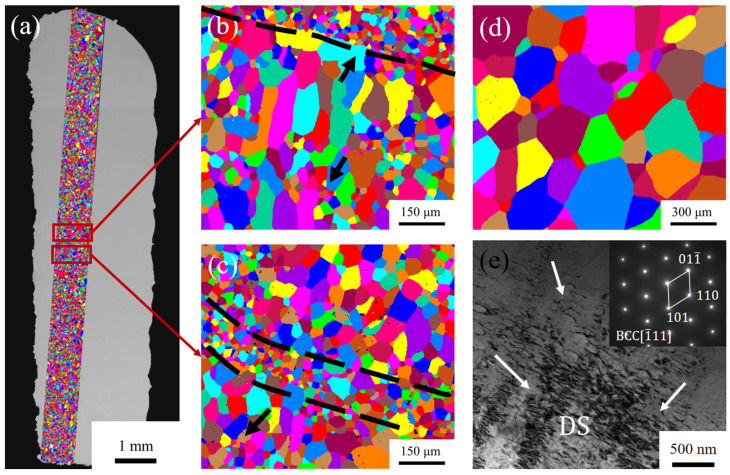
(**a**–**c**) As-deposited and as-cast HEA microstructures by EBSD: global (**a**) and local reconstructions (**b**,**c**) of sedimentary HEA microstructure, (**d**) as-cast HEA, (**e**) BF-TEM images of as-deposited EHA.

**Table 1 materials-16-04039-t001:** Mechanical properties of as-cast and as-deposited Hf_x_NbTa_x_TiZr (x = 0.25) HEA and as-cast Hf_x_NbTa_x_TiZr (x = 1) HEA.

Alloys		σ_s_ (MPa)	R_m_ (MPa)	A (%)
As-deposited	Hf_0.25_NbTa_0.25_TiZr	925 ± 9	960.4 ± 15	28.9 ± 2.3
As-cast	Hf_0.25_NbTa_0.25_TiZr	730 ± 23	756.5 ± 18	20.5 ± 3.8
	HfNbTaTiZr	970 ± 15	972.7 ± 29	22.6 ± 2.1

## Data Availability

Date are contained within the article.

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
