# Peer review of "Composition Design and Tensile Properties of Additive Manufactured Low Density Hf-Nb-Ta-Ti-Zr High Entropy Alloys Based on Atomic Simulations"

_materials, 2023, doi:10.3390/ma16114039_

Round 1

Reviewer 1 Report

In manuscript titled “Composition design and tensile properties of additive manufactured low density Hf-Nb-Ta-Ti-Zr high entropy alloys based on atomic simulations” authors have studied the various properties of HEAs. Based on MD simulations the predicted composition have been experimentally studies. The paper can be accepted for publication after addressing the following points :

1.     Line No.100: How allowing the full relaxation can simulate melting cooling process- Not well explained. Authors should explain it further for the understanding of the reader.

2.     Line No, 105: What is the hypothesis behind allowing only deformation in z- axis

3.     Give more details about identifying the atoms using CNS method

4.     What is rationale of choosing x=0.25 for AM method.? Simulations and experiments show completely opposite trends in the properties. Please clarify

5.     In figure 4 d and 4 e ration is kept constant while varying Ti and Hf. I don’t understand the idea behind this investigation.

Needs to be improved

Author Response

Thank you very much for your overall recognition of our work.We have made corresponding modifications to the issues you raised and made the following explanations and discussions:

1.Line No.100: How allowing the full relaxation can simulate melting cooling process- Not well explained. Authors should explain it further for the understanding of the reader.

Generally, we believe that adding NPT process in AM simulation can reproduce the atomic arrangement of cast alloys in the model. This is because the NPT process provides a certain driving force and time for atomic movement, which further reduces the free energy of atomic arrangement in the final model, just like in the casting process.

Of course, due to the limitations of the model size, this process only replicates the internal situation of the grains during the casting process and cannot be completely equivalent to the casting process, as you pointed out. Therefore, we have made modifications to the expression here, emphasizing only the stability and low free energy of its atomic arrangement form.

2.Line No, 105: What is the hypothesis behind allowing only deformation in z- axis

We do not only allow deformation in the z-direction, but only set the tension value in the z-direction.

3.Give more details about identifying the atoms using CNS method

We were puzzled to find that this article did not use the CNS method. We speculate that you may be referring to common neighbor analysis (CNA). It is the principle used by OVITO software to identify and label atoms. Since CNA is a part of software operation, we have only provided a brief explanation to make readers agree with the reliability of OVITO. If you believe that this part of the content is prone to ambiguity, we will consider deleting this expression.

4.What is rationale of choosing x=0.25 for AM method.? Simulations and experiments show completely opposite trends in the properties. Please clarify

Thank you for your reminder. As mentioned in our introduction, the purpose of our work is to obtain low-density alloys, so we chose x=0.25 with the lowest density for AM forming. Under your reminder, we discovered the oversight in expression and made modifications to line No. 233 by adding corresponding explanations.

As shown in Figure 4(f), the experiment and simulation did not show the opposite trend, but rather were consistent. It was only found through experiments that the content of Hf and Ta elements affects the grain size of the alloy, which cannot be reflected in AM simulation due to the scale limitations of its model. This caused the deviation between the experimental and simulated values.

5.In figure 4 d and 4 e ration is kept constant while varying Ti and Hf. I don’t understand the idea behind this investigation.

We chose to reduce the content of two elements in the same proportion based on the following considerations:

After separately exploring the effects of Hf and Ta elements on the intrinsic properties of the alloy through AM simulation, we found that their effects were opposite, as shown in Figure 4(c).Due to the demand for strength, we have simultaneously reduced the composition of the two elements and hope that their effects can cancel out each other. On the other hand, the purpose of introducing AM simulation in the component design phase is to reduce costs. Reducing the proportion of ingredients can also simplify the subsequent experimental verification process.

Thank you again for your feedback and suggestions. We hope the modifications and explanations we have made are sufficient. Looking forward to the further communication with you.

Reviewer 2 Report

I read your article "Composition design and tensile properties of additive manufactured low density Hf-Nb-Ta-Ti-Zr high entropy alloys based on atomic simulations" which was awarded to Metals – MDPI. In recent years, Additive Manufacturing (AM) technologies have been gaining considerable attention as an innovative method for producing components made of high entropy alloys (HEAs). The unique and excellent mechanical and environmental properties of HEAs can be used in various demanding applications, such as the aerospace and automotive industries.The manuscript is well written and secure. The general intent of this submission is good. The reported experimental methods are presented in sufficient detail and the general characterization techniques are well executed.

Author Response

Thank you very much for your recognition of our work.We will also continue to conduct research in this field and look forward to further communication with you.

Reviewer 3 Report

The paper examines the HEA system of TiZrNbHfTa. By simultaneously reducing the Ta and Hf contents the authors were able to lower the density of the alloy whilst maintaining its high yield strength. LMD was used to refine the grain size, which was higher in the less dense alloy.

The research is good, I would recomend for publication. A few minor comments:

Line:

43: should read ‘widely CONSIDERED and studied’

46: m1/2 needs proper formatting

92-93: sentence should read: ‘The energy of each optimized model was calculate and the lowest one for subsequent MD simulations was selected’

98: Can you give the lattice constant(s)?

281: ‘an amount’ not ‘a amount’. Also, a vague quantity here would be useful, e.g. order of magnitude of the density? An amount sounds very vague.

282: I do not think the simple existence of dislocations proves plasticity. Rather, it is their mobility that proves it.

Conclusions: Whilst you have shown that LMD has reduced the grain size compared with arc melting, I think it would be fair to include a comment that it does come at the cost of a slightly non heterogeneous microstructure.

See above

Author Response

Thank you very much for your overall recognition of our work.We have made corresponding modifications to the issues you raised and made the following explanations and discussions:

43: should read ‘widely CONSIDERED and studied’

46: m1/2 needs proper formatting

92-93: sentence should read: ‘The energy of each optimized model was calculate and the lowest one for subsequent MD simulations was selected’

Thank you very much for discovering and pointing out these issues. We apologize for these omissions in the manuscript and have made revisions to the issues.

98: Can you give the lattice constant(s)?

So glad you pointed out this valuable issue for discussion. To be honest, we are concerned that specific lattice constants may cause some misunderstanding. In the modeling process of LAMMPS, we first established a box composed of Hf atoms with a lattice constant of 4.44. Afterwards, by randomly selecting and replacing Hf atoms with other element atoms, we obtained a model with a lattice constant of 4.44. But this model must undergo the energy minimization treatment we mentioned in the article, which will change the distance between atoms. Therefore, the value of a may vary among different models. We only use the notation in the article to indicate the size of the atoms in the established model more intuitively. I hope this explanation can solve your question.

281: ‘an amount’ not ‘a amount’. Also, a vague quantity here would be useful, e.g. order of magnitude of the density? An amount sounds very vague.

Thank you for pointing out this issue. We have made modifications to the corresponding content.

282: I do not think the simple existence of dislocations proves plasticity. Rather, it is their mobility that proves it.

We fully agree with your viewpoint and have corrected the corresponding expression.

Conclusions: Whilst you have shown that LMD has reduced the grain size compared with arc melting, I think it would be fair to include a comment that it does come at the cost of a slightly non heterogeneous microstructure.

The question you raised is very valuable for discussion. The microstructure of the deposited alloy does indeed exhibit some dimensional non-uniformity, but the anisotropy of strength is not significant in our subsequent tests. At the same time, our team is also trying to use process methods to improve the uniformity of tissue in SLM and LMD samples, and has achieved certain results. We will introduce this section in another article.